# Improving Convergence Speed of Bat Algorithm Using Multiple Pulse Emissions along Multiple Directions

**DOI:** 10.3390/s22239513

**Published:** 2022-12-05

**Authors:** Waqar Younas, Gauhar Ali, Naveed Ahmad, Qamar Abbas, Muhammad Talha Masood, Asim Munir, Mohammed ElAffendi

**Affiliations:** 1Department of Computer Science and Software Engineering, International Islamic University Islamabad, Islamabad 44000, Pakistan; 2EIAS Data Science and Blockchain Lab, College of Computer and Information Sciences, Prince Sultan University, Riyadh 11586, Saudi Arabia

**Keywords:** BAT algorithm, pulse emission, function optimization, levy flights

## Abstract

Metaheuristic algorithms are effectively used in searching some optical solution space. for optical solution. It is basically the type of local search generalization that can provide useful solutions for issues related to optimization. Several benefits are associated with this type of algorithms due to that such algorithms can be better to solve many issues in an effective way. To provide fast and accurate solutions to huge range of complex issues is one main benefit metaheuristic algorithms. Some metaheuristic algorithms are effectively used to classify the problems and BAT Algorithm (BA) is one of them is more popular in use to sort out issues related to optimization of theoretical and realistic. Sometimes BA fails to find global optima and gets stuck in local optima because of the absence of investigation and manipulation. We have improved the BA to boost its local searching ability and diminish the premature problem. An improved equation of search with more necessary information through the search is set for the generation of the solution. Test set of benchmark functions are utilized to verify the proposed method’s performance. The results of simulation showed that proposed methods are best optimal solution as compare to others.

## 1. Introduction

Optimization is considered to be the subset of mathematics; which include review of techniques, procedures, methods, algorithms to obtain optimum result to a given problem [1]. Optimization [2] is process of obtaining best solution of any problem either by using [3] minimization or maximization function, while specifying underline constraints [4] Proposed a new technique to verify the selection of features for using text clustering [5] they describe method to apply feature selection of several data of different time series to discover and apply robotic [6,7]. The research work [8] has described their research f or optimization process that use defined objectives and fitness f unctions. Such fitness functions help to interested parameters and some constraints to give solid solution for issues. Currently optimization has rich scope in everyday life such as business, management or engineering design that reduce the cost, time and resources also to improve the performance, accurate results, good profit [9].

The principle objectives of providing optimized solution are Design variables i.e., a numerical input that will change during the process of optimization; Objective function i.e., describes main motive of the function i.e., either to be minimize or maximize, depending upon nature of problem; Constraints i.e., conditions that must be satisfied while solving the problem and Standard Formulation i.e., representation of problem in mathematical notation. Optimization problems exist in all fields. To solve optimization problems related to engineering disciple which mostly includes designing of hardware components and circuits, planning and scheduling of production, quality controlling, providing main tenancy and repairing of hardware equipment’s [9,10] many optimization methods were proposed in past and proven to be beneficial for solving specific set of problems. Benchmark functions have rich scope in evolutionary optimization field multi programming applications used for optimized solution like stochastic programming applications deals issues of optimizing uncertainty, Hill Climbing programming applications to solve mathematical optimizations issues, constraint programming applications use to identify feasible optimized solution from a large set, Goal based programming application help to find which task are specified to perform, by assigning weights to objectives is use to give value to specific targets, by applying sequential optimization techniques use for selection of multi functions, Gradient based techniques and Linear Programming applications use to find weights and linear function combination. Optimization issues become complex and are increasing frequently, Computational traceability issues occur when global optimal solution is not provided.Despite of computationally extensive and without any guarantee of obtaining optimal solution, metaheuristic approaches are still preferred by many researchers for providing solutions to problems. Though metaheuristic approaches offer many benefits like ease of development and applicable to variety of problems. Even the convergence rate of metaheuristic approaches is better than other optimization approaches [11].

The research work [12] proposed a novel hybrid bat direct search algorithm (HBDS) to resolve integer programming by integrating the bat algorithm with direct search methods. The global modification and the local evolution process were balanced in HBDS. The evolution ability of the proposed algorithm is improved by applying the pattern search method for local searching in place of the random walk method. Finally, the Nelder-Mead technique was utilized to enhance the best solution generated from the bat and pattern search method. The BDS algorithm was studied on 7 integer programming problems and 10 benchmark algorithms of integer programming problem solving to evaluate the performance of the proposed algorithm. The simulated outcomes depicted that HBDS is a capable algorithm and beats the other algorithms in terms of efficiency. Directional echolocation was presented in terms of standard BA in [13] to overcome early convergence, which enhanced the manipulation and investigation capabilities of the standard BA. Three improvements in standard BA refined the performance. The designed method named Directional BA Algorithm (dBA) was evaluated by several functions from a CEC’2005 standard suite. The experimental results from the simulation determined that dBA is superior to others. Ref. [14] the author has applied Standard BA to solve non-linear problems. The proposed algorithm attained better results in comparison to existing metaheuristic techniques.

### 1.1. Problem Statement and Research Significance

The underlying research enhances the bat algorithm’s standard version called multidirectional bat applied and has resulted better than other some algorithms. Multidirectional location optimization is embedded in the bat algorithm to enhance its exploitation and exploration capabilities and consequently significantly enhances the BA performance. The original version of the bat algorithm has two parameters that can be controlled. One is the loudness and another one is the pulse rate. We introduce the values of direction and three other modifications to make an improvement over the original version of the bat algorithm. The lack of exploration capabilities result in a premature convergence of the BA when looking for an optimal solution, which needs to be resolved by introducing a multidirectional exploration.

### 1.2. Research Goals and Objectives

The major goal of this research was to establish an enhanced new bat algorithm that could perform more effectively and efficiently than other bat algorithms. The details of the goals and objectives are as follows:The main goal of this research work was to improve the performance of the bat algorithm to avoid the premature convergence of the BA when searching for an optimal solution, by incorporating algorithmic contributions.The other objective of this research work was to assess the performance of the proposed algorithm with other state-of-the-art algorithms on standard benchmark functions.

### 1.3. Preliminary Study and Pseudocode of the Bat Algorithm

In this section preliminary study about BAT algorithm and Pseudo-code of BAT algorithm are discussed. BAT algorithm has number of parameters that play important role in order to optimize the problem under consideration. Position updation, velocity updation, frequency updation, global position updation, pulse rate updation processes in the BAT algorithm are discussed as under. For instance, observing an unconstrained optimization problem.
(1)min(f(x)),x∈R
where minimization objective function is denoted by $f:R^{n}\to R$ is and a vector x signifies a decision variable. Also, $x\in S\subset R^{n}$ and *S* represents the explore-able space, it is n-dimensional and the parametric constraints are:(2)$$l_{i}\leq x_{i}\leq u_{i}\;i=1,2,3\ldots ,n$$
where $l_{i}$ is the lower bound, $x_{i}$ is location, $x_{i}$. The position updation of BAT’s uses following equation:(3)$$x_{i}\left(t+1\right)=x_{i}\left(t\right)+v_{i}\left(t+1\right)$$
where $x_{i}\left(t+1\right)$ is new position, $x_{i}\left(t\right)$ is old position of ith BAT, $v_{i}\left(t+1\right)$ is the velocity. The velocity updation equation equation is
(4)$$v_{i}\left(t+1\right)=v_{i}\left(t\right)+\left(x_{i}\left(t\right)-p\left(t\right)\right).f_{i}$$
where $v_{i}\left(t\right)$ is old velocity and $x_{i}\left(t\right)$ is position, p(t) is the global optim, $f_{i}$ is the frequency. The frequency can be updated by using following equation
(5)$$f_{i}=f_{min}+\left(f_{max}-f_{min}\right).\beta$$
where $f_{min}$ is the minimum frequency, $f_{max}$ is maximum frequency, β is random amplification factor taken from (0,1). The BA can search globally and locally dependent upon its boundaries; therefore, it is essential to attain an equilibrium among local and global searching abilities by implementing flexible parameters. The local search formula works as follows:(6)$$x_{i}\left(t+1\right)=\vec{p}\left(t\right)+\epsilon A\left(t\right)$$
where ϵ ranges between [−1, 1], $=\vec{p}\left(t\right)$ is global optima and *A*(*t*) is the mean value of loudness of the whole population.

The global searching is accomplished by regulating loudness *A_i_*(*t* + 1) and pulse rate.
(7)$$A_{i}\left(t+1\right)=\alpha A_{i}\left(t\right)$$
where $r_{i}\left(t+1\right)$ and α are parameters with non-zero positive value.
(8)$$r_{i}\left(t+1\right)=r_{i}\left(0\right)\left[1-exp\left(\gamma t\right)\right]$$
where α>0 and γ>0 are constants and $A_{i}\left(0\right)$ and $r_{i}\left(0\right)$ are primary values of loudness and pulse rate, respectively.

The Pseudo-Code of BAT algorithm is given as follows

For every bat i, set the position, velocity, and parameters that arbitrarily yield the frequency using Equation (Equation 5).Calculate and replace the new position and velocity of bat i using Equations (3) and (4).Then, generate rand1 ranging between [0, 1]. If $rand1<r_{i}\left(t\right)$, calculate temporary position and use utility function to calculate fitness value for *i*th bat using Equation (Equation 6).Again, generate another rand2 ranging between [0, 1]. If $rand2<A_{i}\left(t\right)$ and $f\left(x_{i}\left(t\right)\right)<f\left(p\left(t\right)\right)$, then replace $A_{i}\left(t\right)$ and $r_{i}\left(t\right)$ using values from Equations (7) and (8), respectively.Sort all positions according to their fitness values and extract the optimum position.Then move on to Step 2 if the stopping criteria is not reached.

## 2. Literature Review

In 2010 a hug research work was done in the field of BAT algorithm and this era has large application of BAT algorithm due to that performance and implementation of BAT algorithm has become efficient.The presented algorithm attained better results in comparison to existing metaheuristic techniques. Meng et al. [15] introduced doppler effect in original BA. Author used Rechenberg’s one fifth mutation rule and Gaussian probabilities function to enhance the ability of echolocation system. The author reported that their technique amplified movement of BA for searching global solution. In modified BA, author reformed the equation to calculate loudness and pulse emission rate for each bat. The modified bat algorithm (MBA) was tested on 15 benchmark functions and reported results were improved over standard BA. Huang et al. [16] reformed the procedure of local searching by presenting the Gaussian walk concept for bats. The author transformed the velocity update equation of standard BA which resulted in higher diversity in population. The searching dimensions in BA have surged by this method. The Cloud model BA (CBA) which merges the concept of cloud model into BA was introduced in 15 [17]. Cloud model is famous with its exceptional features of presenting abstruse knowledge. The author modified the echolocation model of BA with the help of transformation theory of cloud model. Also, developed an algorithm which showed better results on optimizations function. Compact bat algorithm (cBA) presented in [18] was studied for environments which had limited resources related to hardware. The design variable of search space solution was reformed with a probability-based population model. The author reported that this method can be implemented in limited memory environments. A mutation based unique bat algorithm with the help of image processing was presented by waqas et al. [19]. Two alterations were introduced in standard BA. First, frequency and loudness were considered to be fixed and second, for diversity of population, a mutation operator was developed. Tests were carried on image processing-based examples and reported that this technique provided better result as compared with original BA. This hybridization technique resulted in increasing the capacity of local search. Chakri et al. [14] enhanced the speed of convergence in BA by introducing the opposition- based numbering concept. The author simulated the revised algorithm for various benchmark functions. It was observed that the approach resulted in better accuracy as well as convergence speed for global search. The author in [20] enhanced the efficiency of standard BA in respect of convergence rate and accuracy. The concept of levy flights trajectory was presented which improved the diversity of population which resulted in making the solution to evacuate from local minima. A differential operator was also utilized to improve the convergence speed. The author simulated various scenarios on well-known benchmark functions and reported that their method can estimate better in high dimensional space. Garip et al. [21] implemented chaotic sequence for parameter adjustment in standard BA known as Chaotic BA (CBA). The author presented the impact of different chaotic sequences on convergence performance of standard BA. Simulation of the algorithm resulted in better performance of CBA over standard BA. Osaba et al. [22] proposed an improved Bat Algorithm called (DalBat). The author considered Hamming Distance (HC) to calculate the distance from one bat to another. Moreover, an additional method was implemented in the algorithm which depends on two structures of the neighborhood bats, the optimum searching agent in the population finds them. The author also stated that they considered the bat algorithm to resolve a problem from bioinformatics for the first time. Hong et al. [23] proposed a novel procedure to compute the motion of the floating platform. A vector regression method was considered to simulate the movement of a floating platform and a novel BA was proposed to improve the operators of the method. Moreover, an empirical mode decomposition (EMD) method was used to decompose the time series signal. The author also proved the reliability and activity of the proposed method. Primary issues of bat algorithm related to low convergence rate and trapping in local optima intrigued in [21] to enhance the performance of bat algorithm in terms of exploitation by using a variable dimension size. This process assisted to choose random values for the subsequent epoch and the chosen value overlooked representing impractical dimension region. The modified algorithm was validated on ten benchmark functions and the results showed that it performed better in terms of optimization.The research work [22] presented a Principal Component Analysis (PCA) based bat algorithm called PCA-BA. Every searching agent is utilized when PCA is applied. It excludes the uncorrelated and the individuals in the population below threshold of similarity. The algorithm was tested on CEC’2008 benchmark functions and the results revealed that PCA-BA performed exceptionally well against basic bat algorithm. Moreover, the algorithm was also compared with other algorithms from literature in terms of optimization and showed better performance.Wang et al. in [24] considered eight selection approach based on fitness function value to proposed a unique algorithm named as MixBAT. The algorithms performance was validated using CEC’2013 benchmark functions and comparison with some other methods from literature. MixBAT attains the global optimum without being trapped in local optima. But, occasionally it gets trapped in a local optimum when other approaches could be invalid in the initial iterations. There are two constraints of innovative form of the Bat Algorithm that can be well-ordered. First one is s loudness and the second is as rate of pulse. Chakri et al. [14] strained direction values and some changes to improve the novel version of BAT algorithm. Lack of exploration capabilities results in premature convergence and minimized accuracy of BAt in finding optimal solution which is needed to be resolved by introducing multi-directional exploration.

## 3. Proposed Enhancement

An algorithm based on the multiple directions is used to solve a generic optimization algorithm with capability of parallel computation properties. The multi-directions-based algorithm is a direct searching algorithm. Out of many benefits, the multiple directions-based searching is assisted by convergence theorems which are numerically tested to perform optimization making this algorithm superior over the others. Given that the method is assisted by convergence theorems, the algorithm will also result in better performance on high dimensional problems.

### 3.1. Pseudo Code of Multi Directional Searching Algorithm

The summary of key steps involved in multiple directions-based algorithm as follows:In the beginning the values of the expansion factor (μ), contraction factor (θ) and the maximum iterations count ($M_{itr}$) are assigned.The algorithm starts with a simplex ($S_{o}$) with vertices $x^{o}$, where i = 0, 1, 2, … n.The assortment of vertices in ascending order is done on the basis of their functions.(9)$$f\left(X_{0}\right)\leq f\left(X_{0}\right)\ldots \leq f\left(X_{0}\right)$$The main loop is started with epoch 1 until the maximum number of iterations are reached.The vertices $x_{1}$, $x_{2}$, …, $x_{n}$ are used for evaluation through the best vertex *X*_0_ until the new values are attained.(10)$$x_{i}=X_{0}+a_{i}\left(x_{1}-X_{0}\right)+\ldots +z_{i}\left(x_{n}-X_{0}\right)$$(11)$$f_{x}i=f(x_{i})$$(12)$$f_{x_{\phi }}=min_{i}\left\{f_{x_{i}}\right\}$$where i=1,… population_size and $a_{i}\ldots z_{i}$ are constant values to keep the solution within search space. f(. ) is an evaluation function.If the new value of a vertex gives better results than the current best vertex, then the algorithm begins with the expansion process.The expansion procedure begins to increase each edge by considering μ, where μ=2 to generate new increased vertices. The new increased vertices are assessed to validate the achievement of the expansion process.If the increased vertex is superior over the remaining vertices, the new simplex will be the increased simplex.If the expansion process fails to deliver the expected outcome, then the contracted simplex begins to operate by altering the step size via with the help of θ.The assortment of new vertices according to the values from the respective evaluation functions and the new simplex is created.The iteration number is added and the procedures are repeated until $M_{itr}$. Finally, the best solution is achieved.

### 3.2. Parameter Setting

The advancements of latest technology have admonished researchers to modify the methods of optimization. But it is not mean to use new methods from scratch but do some changes in the methods of optimization that are available so results can be more efficient and effective. The results of the proposed methodology make a comparison with the prior technique. 19 standard optimization functions were selected to check the performance of the proposed algorithm. Most of the research work in the literature has adopted a population size of 20. Hence, for both simulations, a population size of 20 was considered and the number of iterations was selected to be 500 to ensure fairness and comparability. Also, other parameters, α, 266 and y were chosen to be at 0.9, initial loudness A0 to be 0.25, and pulse emission rate r0 to be 267 0.5. The simulation was carried out on MATLAB 2018a with operating system Windows10 and memory size 4 GB on Intel(R) Core (TM) i5-2450M CPU @ 2.50 GHz 2.50 GHz.

### 3.3. Results

Table 1 describes the proposed approach evaluation on the basis of statistical results. The proposed technique was compared with three other algorithms by running the functions given in the previous section 10 times separately. The dimension for each algorithm was set to 20, 30 and 100 to evaluate the algorithms. Four performance measures, the best, average, worst and standard deviation of the optimal value, were obtained and these results are shown in Table 1.

The significance of an algorithm in terms of performance can be viewed clearly on a convergence curve. It is used to observe the ability of an algorithm to escape from local optima and its convergence speed. Further, the results of simulating the proposed algorithm and the other techniques using a dimension of 30 on benchmark functions are shown in Figure 1, Figure 2, Figure 3, Figure 4, Figure 5, Figure 6, Figure 7, Figure 8, Figure 9 and Figure 10.

The convergence graph of Salomon’s function showed that the proposed algorithm performed better than all the other algorithms. The convergence results of the sum of different power functions highlighted that the proposed algorithm performed better than the other algorithms and proved its effectiveness in terms of finding global optima and increasing convergence speed. The multimodal functions are known for having multiple local optima, which makes finding global optima complicated.

The convergence graphs of the rotated hyperellipsoid function and Griewank function showed that the proposed algorithm performed better than the other algorithms and proved its effectiveness in terms of finding global optima and increasing convergence speed. The multimodal functions are known for having multiple local optima, which makes finding global optima complicated.

The convergence graphs of the Trid function and Rastrigin function showed that the proposed algorithm performed better than the other algorithms and proved its effectiveness in terms of finding global optima and increasing convergence speed. The multimodal functions are known for having multiple local optima, which makes finding global optima complicated.

The convergence graphs of the Lévy function and Ackley function showed that the proposed algorithm performed better than the other algorithms and proved its effectiveness in terms of finding global optima and increasing convergence speed. The multimodal functions are known for having multiple local optima, which makes finding global optima complicated.

The convergence graphs of the Schwefel function and Rosenbrock function demonstrated that the proposed algorithm performed better than the other algorithms and proved its effectiveness in terms of finding global optima and increasing convergence speed. The multimodal functions are known for having multiple local optima, which makes finding global optima complicated.

The convergence graphs of the Schaffer function and Styblinski–Tang function showed that the proposed algorithm performed better than the other algorithms and proved its effectiveness in terms of finding global optima and increasing convergence speed. The multimodal functions are known for having multiple local optima, which makes finding global optima complicated.

The convergence graphs of the Weierstrass function and Zakharov function highlighted that the proposed algorithm performed better than other algorithms and proved its effectiveness in terms of finding global optima and increasing convergence speed. The multimodal functions are known for having multiple local optima, which makes finding global optima complicated.

Alpine (o) and Bent- cigar (p) convergence graphs showed that the proposed algorithm performed better than the other algorithms and proved its effectiveness in terms of finding global optima and increasing convergence speed.

The convergence graphs of the Dixon–Price (q) and Michalewicz functions (r) demonstrated that the proposed algorithm performed better than other algorithms and proved its effectiveness in terms of finding global optima and increasing convergence speed. The multimodal functions are known for having multiple local optima, which makes finding global optima complicated.

The convergence graph of the Powell function (s) showed that the presented method worked as effectively as the other methods and proved its effectiveness and increasing speed of convergence. The multimodal functions are known for having multiple local optima, which makes finding global optima complicated.

### 3.4. Summary of Results

Unimodal functions are usually considered to observe the ability of an algorithm to escape local optima and to assess its convergence speed. While looking at Figure 3, Figure 4, Figure 5 and Figure 6 and Figure 10 above, the proposed algorithm performed better than the other algorithms and proved its effectiveness in terms of finding global optima and increasing convergence speed. The multimodal functions are known for having multiple local optima, which makes finding global optima complicated. We can see in Figure 1, Figure 2 and Figure 7, Figure 8 and Figure 9 depicting the results of multimodal functions that the proposed algorithm performed better. In short, the proposed algorithm showed better efficiency in terms of optimizing any of the standard functions. Finally, the convergence speed of each algorithm used in the experiment was evaluated in Table 1. The values given in the table represented the iteration number at which each algorithm converged to its minimum value. The results showed that the proposed algorithm outperformed all other algorithms in terms of convergence speed. These values were obtained from the simulation experiment with a dimension of 30 and the convergence speed comparison between the proposed and the other three algorithms by varying the dimensions and population size.

## 4. Statistical Significance

This section shows the statistical significance of the proposed algorithm and two state-of-the-art algorithms. The statistical significance of the proposed algorithm and mixBA algorithm is reported in Table 2 of this paper. It can be observed from Table 2 that the *p*-value (0.000140339) of a two-tailed test was less than the level of significance (0.05), which showed that there was a significant improvement in the performance of the proposed algorithm over the mixBA algorithm. The degree of freedom was 18 and the Pearson correlation was 0.45715803. Table 3.

The statistical significance of the proposed algorithm and EBA algorithm is reported in Table 3 of this paper. It can be observed from Table 3 that the *p*-value (0.005000611) of a two-tailed test was less than the level of significance (0.05), which showed that there was a significant improvement in the performance of the proposed algorithm over the EBA algorithm.

## 5. Conclusions

The major aim of this research was to enhance the bat algorithm’s standard version by introducing a multidirectional bat algorithm, which resulted in better performance than some other algorithms. In this research, the selection of an optimal target depended upon the range between the target and the bat and the direction in which the target was moving. In the proposed strategy, the distance along multiple directions was calculated to estimate the range between the target and the bat. The proposed algorithm was tested on standard benchmark functions in search of an optimal value. The experimental results were generated using parameter settings given in Section 4 of this paper. The results of the average fitness were reported for the proposed algorithm and two other standard state-of-the-art algorithms, mixBA and EBA, for all functions. The proposed algorithm was helpful in escaping from local optima and showed a dominating performance compared with the mixBA and EBA algorithms. We used the best, worst, average and standard deviation of the average fitness results as shown in Table 1 of this paper. It can be concluded from the experimental results that the proposed algorithm has better performance than the other algorithms. The proposed technique can get out of local minimum and can solve the optimization issues in an effective and efficient way. Future work will memorize the convergence track and then utilize this information to further enhance the convergence speed of the bat algorithm. 

## Figures and Tables

**Figure 1 sensors-22-09513-f001:**
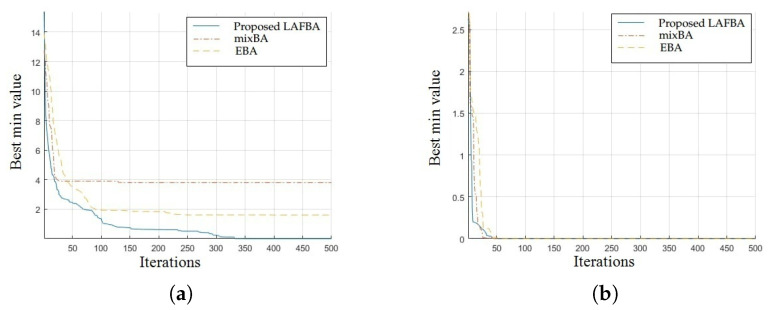
(**a**) Salomon’s function convergence graph. (**b**) Sum of different power functions.

**Figure 2 sensors-22-09513-f002:**
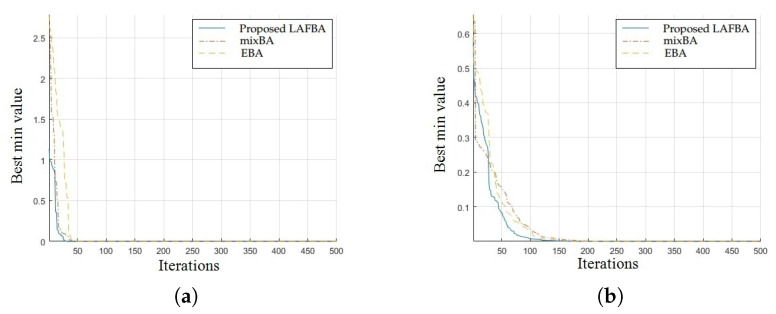
(**a**) Rotated hyperellipsoid convergence graph. (**b**) Griewank function convergence graph.

**Figure 3 sensors-22-09513-f003:**
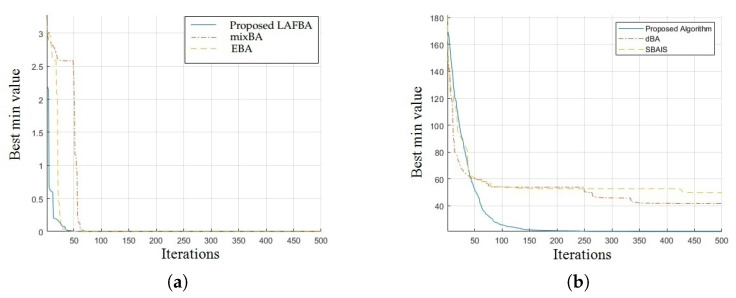
(**a**) Trid function convergence graph. (**b**) Rastrigin function convergence graph.

**Figure 4 sensors-22-09513-f004:**
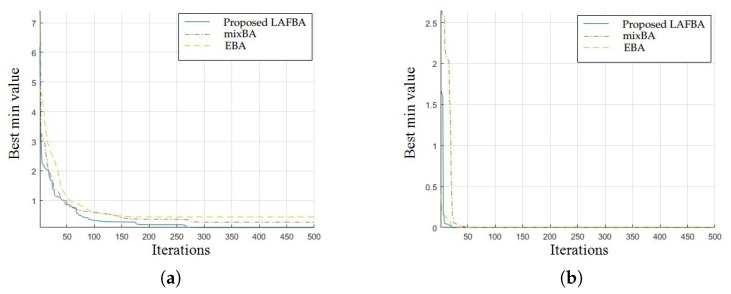
(**a**) Lévy function convergence graph. (**b**) Ackley function convergence graph.

**Figure 5 sensors-22-09513-f005:**
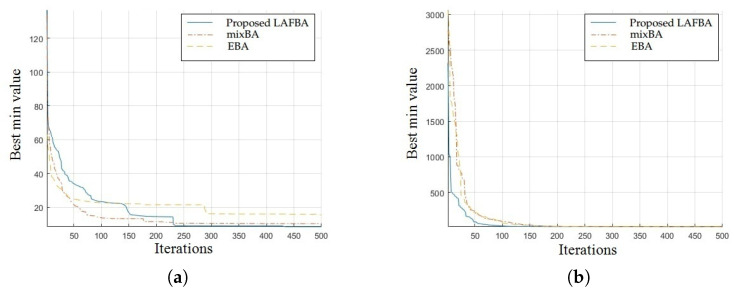
(**a**) Schwefel function convergence graph. (**b**) Rosenbrock function convergence graph.

**Figure 6 sensors-22-09513-f006:**
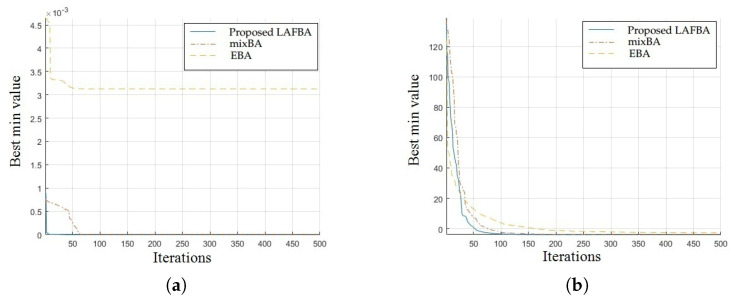
(**a**) Schaffer function convergence graph. (**b**) Styblinski–Tang function.

**Figure 7 sensors-22-09513-f007:**
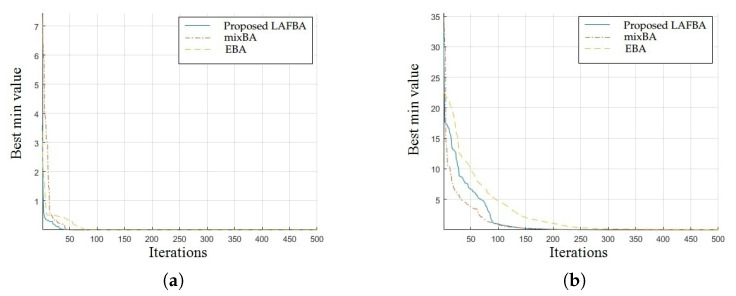
(**a**) Weierstrass function convergence graph. (**b**) Zakharov function convergence graph.

**Figure 8 sensors-22-09513-f008:**
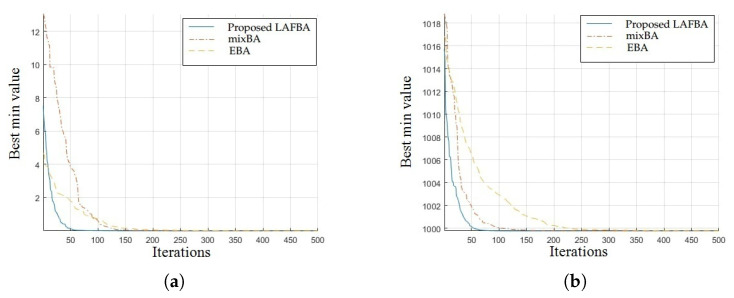
(**a**) Alpine function convergence graph. (**b**) Bent-cigar function convergence graph.

**Figure 9 sensors-22-09513-f009:**
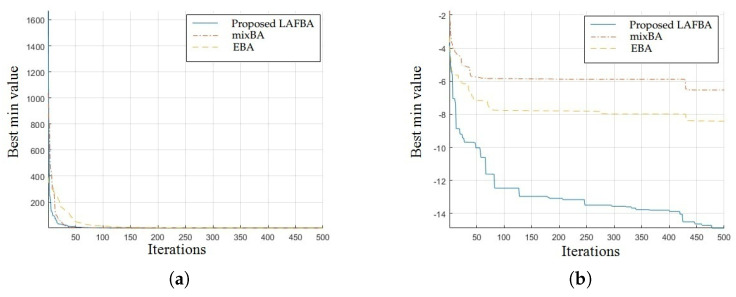
(**a**) Dixon–Price function convergence graph. (**b**) Michalewicz function convergence graph.

**Figure 10 sensors-22-09513-f010:**
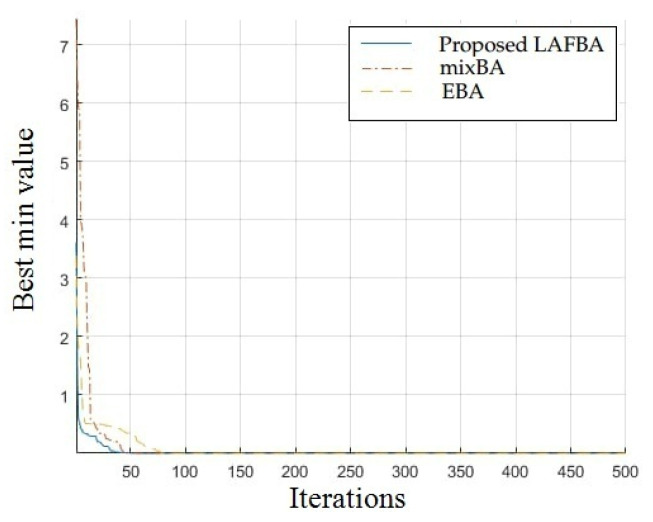
Powell function convergence graph.

**Table 1 sensors-22-09513-t001:** Comparison of average fitness values of proposed LAFBA algorithm with mixBA and EBA algorithms.

Function	Algorithm	Best	Average	Worst	Standard Deviation
F_S_	Proposed LAFBA	1.34 × 10^−2^	1.30 × 10^−2^	2.18 × 10^−2^	3.61 × 10^−3^
	mixBA	1.41 × 10^−3^	0.98 × 10^−2^	1.27 × 10^−2^	2.13 × 10^−1^
	EBA	2.41 × 10^−2^	1.01 × 10^−1^	2.10 × 10^−2^	1.81 × 10^−8^
F_FDP_	Proposed LAFBA	1.32 × 10^−2^	1.23 × 10^−2^	1.51 × 10^−2^	2.83 × 10^−1^
	mixBA	2.96 × 10^−2^	4.09 × 10^−2^	1.24 × 10^−2^	2.30 × 10^−1^
	EBA	3.18 × 10^−2^	4.71 × 10^−1^	2.30 × 10^−1^	3.98 × 10^−0^
F_RHE_	Proposed LAFBA	2.54 × 10^−3^	3.04 × 10^−2^	3.09 × 10^−2^	3.94 × 10^−1^
	mixBA	3.61 × 10^−2^	2.82 × 10^−1^	4.13 × 10^−1^	5.38 × 10^−1^
	EBA	2.45 × 10^−2^	7.01 × 10^−1^	2.98 × 10^−1^	4.05 × 10^−1^
F_G_	Proposed LAFBA	2.13 × 10^−3^	1.93 × 10^−3^	1.93 × 10^−1^	1.36 × 10^−1^
	mixBA	3.44 × 10^−2^	4.17 × 10^−1^	2.71 × 10^−1^	1.07 × 10^−1^
	EBA	2.96 × 10^−2^	5.24 × 10^−1^	1.52 × 10^−1^	4.19 × 10^−1^
F_T_	Proposed LAFBA	1.53 × 10^−2^	3.03 × 10^−2^	4.18 × 10^−1^	2.45 × 10^−1^
	mixBA	4.74 × 10^−2^	4.95 × 10^−1^	6.34 × 10^−1^	3.06 × 10^−1^
	EBA	3.22 × 10^−1^	5.05 × 10^−1^	4.48 × 10^−1^	1.84 × 10^−1^
F_R_	Proposed LAFBA	3.02 × 10^−2^	4.094 × 10^−2^	5.16 × 10^−1^	1.700 × 10^−1^
	mixBA	1.41 × 10^−1^	1.82 × 10^−1^	3.48 × 10^−1^	5.82 × 10^−1^
	EBA	5.01 × 10^−1^	7.03 × 10^−1^	2.71 × 10^−1^	1.59 × 10^−0^
F_L_	Proposed LAFBA	2.76 × 10^−3^	2.14 × 10^−2^	4.51 × 10^−1^	3.49 × 10^−1^
	mixBA	1.84 × 10^−2^	9.23 × 10^−1^	8.64 × 10^−1^	9.79 × 10^−1^
	EBA	6.73 × 10^−1^	5.63 × 10^−1^	2.55 × 10^−1^	3.10 × 10^−1^
F_A_	Proposed LAFBA	1.41 × 10^−3^	1.87 × 10^−2^	1.41 × 10^−1^	7.63 × 10^−1^
	mixBA	8.36 × 10^−2^	4.90 × 10^−2^	6.128 × 10^−1^	4.16 × 10^−1^
	EBA	2.04 × 10^−2^	4.67 × 10^−2^	1.41 × 10^−1^	8.04 × 10^−1^
F_SC_	Proposed LAFBA	1.16 × 10^−2^	1.31 × 10^−1^	4.037 × 10^−1^	3.56 × 10^−0^
	mixBA	4.61 × 10^−2^	5.04 × 10^−1^	1.52 × 10^−1^	2.47 × 10^−0^
	EBA	4.46 × 10^−2^	4.20 × 10^−1^	2.41 × 10^−1^	1.17 × 10^−0^
F_RB_	Proposed LAFBA	2.83 × 10^−2^	1.09 × 10^−1^	1.11 × 10^−1^	2.96 × 10^−0^
	mixBA	3.91 × 10^−2^	8.21 × 10^−1^	5.74 × 10^−1^	5.04 × 10^−0^
	EBA	8.54 × 10^−2^	5.14 × 10^−1^	1.50 × 10^−1^	6.90 × 10^−0^
F_SCH_	Proposed LAFBA	1.32 × 10^−2^	1.23 × 10^−2^	1.51 × 10^−2^	2.83 × 10^−1^
	mixBA	2.16 × 10^−2^	4.09 × 10^−2^	1.24 × 10^−2^	2.30 × 10^−1^
	EBA	3.18 × 10^−2^	4.71 × 10^−1^	2.30 × 10^−1^	3.98 × 10^−0^
F_ST_	Proposed LAFBA	2.54 × 10^−3^	3.04 × 10^−2^	3.09 × 10^−2^	3.94 × 10^−1^
	mixBA	3.61 × 10^−2^	2.82 × 10^−1^	4.31 × 10^−1^	5.38 × 10^−1^
	EBA	2.17 × 10^−2^	7.31 × 10^−1^	2.88 × 10^−1^	4.05 × 10^−1^
F_W_	Proposed LAFBA	2.13 × 10^−3^	1.93 × 10^−3^	1.93 × 10^−1^	1.36 × 10^−1^
	mixBA	3.44 × 10^−2^	4.71 × 10^−1^	2.71 × 10^−1^	1.07 × 10^−1^
	EBA	2.96 × 10^−2^	5.24 × 10^−1^	1.52 × 10^−1^	4.19 × 10^−1^
F_Z_	Proposed LAFBA	1.53 × 10^−2^	3.03 × 10^−2^	4.18 × 10^−1^	2.45 × 10^−1^
	mixBA	4.74 × 10^−2^	4.95 × 10^−1^	6.34 × 10^−1^	3.06 × 10^−1^
	EBA	3.22 × 10^−2^	5.05 × 10^−1^	4.48 × 10^−1^	1.84 × 10^−1^
F_A_	Proposed LAFBA	3.02 × 10^−2^	4.094 × 10^−2^	5.16 × 10^−1^	1.700 × 10^−1^
	mixBA	1.41 × 10^−1^	1.82 × 10^−1^	3.48 × 10^−1^	5.82 × 10^−1^
	EBA	5.01 × 10^−1^	7.03 × 10^−1^	2.71 × 10^−1^	1.59 × 10^−0^
F_B_	Proposed LAFBA	2.76 × 10^−3^	2.14 × 10^−2^	4.51 × 10^−1^	3.49 × 10^−1^
	mixBA	1.84 × 10^−2^	9.23 × 10^−1^	8.64 × 10^−1^	9.79 × 10^−1^
	EBA	6.73 × 10^−1^	5.63 × 10^−1^	2.55 × 10^−1^	3.10 × 10^−1^
F_D_	Proposed LAFBA	1.41 × 10^−3^	1.87 × 10^−2^	1.41 × 10^−1^	7.63 × 10^−1^
	mixBA	8.36 × 10^−2^	4.90 × 10^−2^	6.128 × 10^−1^	4.16 × 10^−1^
	EBA	2.04 × 10^−2^	4.67 × 10^−2^	1.41 × 10^−1^	8.04 × 10^−1^
F_M_	Proposed LAFBA	1.16 × 10^−2^	1.31 × 10^−1^	4.037 × 10^−1^	3.56 × 10^−0^
	mixBA	4.61 × 10^−2^	5.04 × 10^−1^	1.52 × 10^−1^	2.47 × 10^−0^
	EBA	4.46 × 10^−2^	4.20 × 10^−1^	2.41 × 10^−1^	1.17 × 10^−0^
F_P_	Proposed LAFBA	2.83 × 10^−2^	1.09 × 10^−1^	1.11 × 10^−1^	2.96 × 10^−0^
	mixBA	3.91 × 10^−2^	8.21 × 10^−1^	5.74 × 10^−1^	5.04 × 10^−0^
	EBA	8.54 × 10^−2^	5.14 × 10^−1^	1.50 × 10^−1^	6.90 × 10^−0^

**Table 2 sensors-22-09513-t002:** Statistical significance of proposed algorithm and mixBA algorithm.

	Proposed Algorithm	mixBA
Mean	0.012014737	0.049726842
Variance	0.000110537	0.001419668
Observations	19	
Pearson correlation	0.45715803	
Hypothesized mean difference	0	
df	18	
t stat	−4.809881796	
P(T ≤ t) two-tail	0.000140339	
t critical two-tail	2.100922037	

**Table 3 sensors-22-09513-t003:** Statistical significance of proposed algorithm and EBA algorithm.

	Proposed Algorithm	EBA
Mean	12.01473684	183.4684211
Variance	110.5369041	55583.28561
Observations	19	
Pearson correlation	0.207952704	
Hypothesized mean difference	0	
df	18	
t stat	−3.196518588	
P(T ≤ t) two-tail	0.005000611	
t critical two-tail	2.100922037	

## Data Availability

Not applicable.

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
