# Peer review of "Improving Convergence Speed of Bat Algorithm Using Multiple Pulse Emissions along Multiple Directions"

_sensors, 2022, doi:10.3390/s22239513_

Round 1
Reviewer 1 Report
The subject of this paper is interesting. However, needs moderate revision.
1. In section 1.0: Line numbers: 36, 37, 39, and 45; check them very carefully.
2. In section 2.0: Line no: 70; check it very carefully.
3. Novelty of the work should be established.
4. What about fig. 3-1 (flowchart) and fig. 4-1?
5. The authors should include the figure captions properly.
6. Rewrite the table 1 caption.
7. Proposed solution should be compared with previously reported algorithms. Explain how is your proposed solution algorithm more novelty.
8. Figures are not clear; check them very carefully.
9. Typo errors in the manuscript. Check it very carefully.
10. The authors should include the acknowledgment part, author contribution section, etc.
Reviewer 2 Report
The goals and objectives of the study should be more clearly defined. It is also necessary to display the practical significance of this study.
Reviewer 3 Report
Dear authors, while the paper is nice in shape, there are a few comments and/or suggestions to improve the manuscript. There is some interest in this type of research, but I found this paper only mildly interesting in its present form. Please strongly consider the following suggestions:
- The significance of the study is not clear to me and there is a serious literature review gap in this paper. I strongly consider that the distinguished authors did not pay attention to the relevant cited references. In this context, please clarify better the advantages of this paper in the introduction section because in the literature a lot of recent papers consider the same proposed approach. I strongly recommend the authors reconsider the related work section for the literature review and discuss the drawbacks of existing works.
- A lot of references are missing [8, 10, 11, 21, 22, 23, etc.].
- Please describe the necessity of each reference in the first section, because multiple citations contradict the ethics of journals with international visibility. Please also reconsider [1-7].
- Also, the authors must revise the article sections. The research is not conducted correctly. Section two must be integrated, and all the references must be consecutively used.
- The authors use the BAT algorithm. Please emphasize the BA parameter values, loudness, and pulse rate. Also, all the BA parameters are missing.
- The methodology itself is not well described. There are some simple mathematical quantities, but their use in the proposed technique is not precisely defined.
- The results analysis is not enough. I cannot see deeply analysis related to them and cannot understand the meaning of the results. Please add more analysis. The readers would benefit from a more insightful discussion of the results and a clear statement about the main conclusions drawn from the research carried out.
- In the conclusion, the section starts with a brief explanation of the paper's goal and explains what the significant findings are and why your paper is really important.
In this form, the paper can be rejected. The authors must revise the whole manuscript.
Round 2
Reviewer 3 Report
In this form, the paper can be publiched!